# LeLaN: Learning A Language-Conditioned Navigation Policy from In-the-Wild Videos

**Noriaki Hirose**[1,2], **Catherine Glossop**[1†], **Ajay Sridhar**[1†], **Oier Mees**[1], **Sergey Levine**[1]

[1] University of California Berkeley    [2] Toyota Motor North America    [†] core contributors

noriaki.hirose@berkeley.edu

**Abstract:** The world is filled with a wide variety of objects. For robots to be useful, they need the ability to find arbitrary objects described by people. In this paper, we present LeLaN (Learning Language-conditioned Navigation policy), a novel approach that consumes unlabeled, action-free egocentric data to learn scalable, language-conditioned object navigation. Our framework, LeLaN leverages the semantic knowledge of large vision-language models, as well as robotic foundation models, to label in-the-wild data from a variety of indoor and outdoor environments. We label over 130 hours of data collected in real-world indoor and outdoor environments, including robot observations, YouTube video tours, and human walking data. Extensive experiments with over 1000 real-world trials show that our approach enables training a policy from unlabeled action-free videos that outperforms state-of-the-art robot navigation methods, while being capable of inference at 4 times their speed on edge compute. We open-source our models, datasets and provide supplementary videos on our project page [1].

**Keywords:** Language-conditioned navigation, Data augmentation

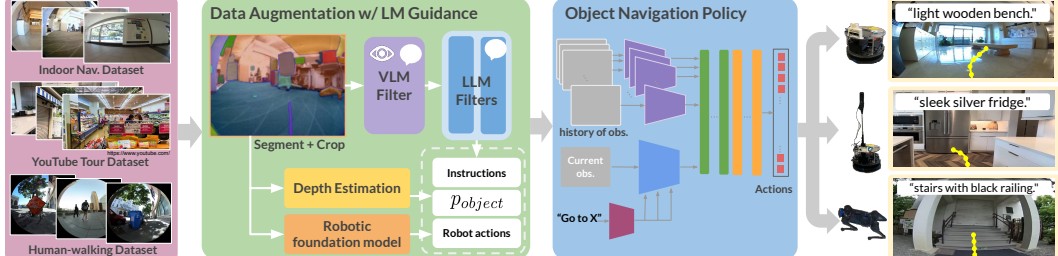

Figure 1: **LeLaN** leverages foundation models to label in-the-wild video data with language instructions for object navigation. We train a state-of-the-art robot policy on this data for solving challenging zero-shot language-conditioned object navigation tasks across a variety of indoor and outdoor environments.

## 1 Introduction

One of the main challenges in robotics is enabling robots to interpret and execute arbitrary user-specified language instructions. For instance, in the case of navigation, a mobile robot should understand that "go to the white and grey chair" and "go to the office chair next to the white desk and black computer monitor" could be asking the same thing in different ways. A major bottleneck of training language-conditioned control policies is access to high-quality and diverse language- and action-annotated robotics datasets, as acquiring human annotations for such datasets is an expensive process. Some large datasets exist for training language-conditioned manipulation policies [1, 2, 3, 4], but minimal data exists for training language-conditioned navigation policies. Additionally, existing language-annotated robotics datasets are often not large enough to train end-to-end language-conditioned policies that exhibit strong generalization capabilities akin to foundation models in domains such as natural language processing and computer vision. [5, 6, 7].

---

[1]**https://learning-language-navigation.github.io/**

8th Conference on Robot Learning (CoRL 2024), Munich, Germany.

Past works try to circumvent the lack of language-annotated robotics data by applying foundation models, such as vision language models (VLMs) and large language models (LLMs) in a zero-shot manner. This includes higher-level planning [8, 9, 10, 11, 12, 13], code-generated policies [14, 15, 16, 17], and zero-shot control [18]. The issue these methods face is that the data distribution used to train these foundation models is vastly different from the distribution of embodied robot experience. Additionally, these foundation models are often too computationally expensive to run on edge, which is important for mobile robotics. Other methods try to fine-tune these models on a more narrow distribution of in-domain robotics data [19]. However, it is unclear if these model retain their full generalization capabilities after fine-tuning. In this paper, we approach the problem of grounding the abilities of foundation models in embodied experience in the context of language-conditioned policies for visual navigation by using scalable data augmentation.

In this paper, we leverage foundation models for vision, language, and robot actions, to annotate egocentric navigation experience with *physically grounded* language and action labels. Our dataset consists of egocentric navigation data from indoor ground robots, first-person view (FPV) tours on YouTube, and FPV data of a person walking in indoor and outdoor environments. We segment each image in the dataset with a segmentation model to retrieve objects of interest for our language instructions. We use a VLM to generate a description of the objects and an LLM to produce various language instructions for object navigation. We then generate counterfactual action labels that navigate the robot to the object in question using a robot foundation model (RFM) for navigation [20] and 3D estimated point cloud information, which is generated from a monocular depth estimation model [21, 22]. The complete process can be fully automated on any egocentric data and the resulting augmented dataset combines the strengths of spatially- and semantically-aware foundation models. Policies trained on this data demonstrate broad generalization and robust physical grounding for language-conditioned navigation. Our contributions are the following:

1. A general data labeling method guided by foundation models that can consume unlabeled videos of action-free data,
2. Empirical results showing that the produced labels enable training a state-of-the-art policy that is more robust for noisy instructions, dynamic target object, and obstacles avoidance,
3. Data-level ablations showing the importance of including in-domain robotics data to ground our trained policies and the effectiveness of our data augmentation,
4. We open source over 120 hours of egocentric data annotated with language and actions, including over 15 hours of human-collected video from 11 cities in 3 countries.

## 2  Related Works

**Language-Conditioned Policies for Robotics:** Learning end-to-end language-conditioned policies for robotics has been studied extensively [23, 24, 19, 25, 26, 27, 28]. Learned methods are appealing because of their ability to generalize beyond objects and tasks not seen in the training data. However, they often require language- and action-annotated data, which can be expensive to collect and annotate in the real world. It may be cheaper to automate data collection in simulation [16, 29, 30], but policies trained in simulation often transfer poorly to unstructured real-world environments, which is commonly referred to as the sim2real gap [31, 32].

**Language-Drive Zero-Shot Object Navigation:** In this paper, we explore the problem of language-driven zero-shot object navigation (L-ZSON) [5], where an agent navigates toward an object specified by a language command. The language input is often unstructured and could refer to an object that the agent has never seen before. Several methods attempt to apply foundation models zero-shot for language-guided vision-based navigation [12, 33]. However, the primary baseline we consider in our experiments is CLIP on Wheels (CoW) [34], which uses CLIP, a multi-modal encoder that embeds text and images in a shared latent space. CoW uses a CLIP [5] image encoder to compute similarity scores with a CLIP text embedding corresponding to the target object. Unlike many previous ZSON works, which also evaluate search efficiency [35, 34, 36, 37], we focus on the case where the object is within the line of sight of the robot [38]. This is because our system could be easily

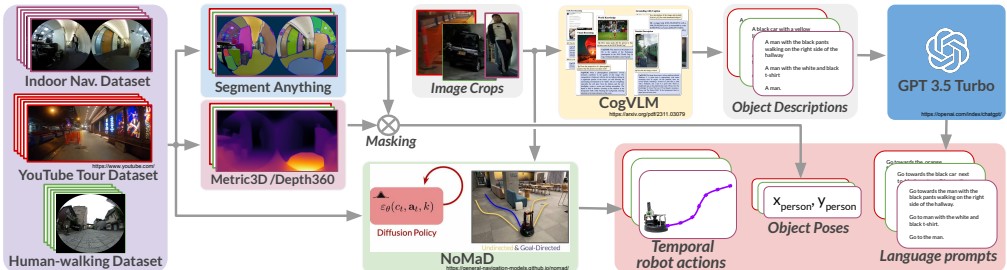

Figure 2: **Data Annotation.** To generate diverse language instructions for object navigation, we pass generic egocentric image observations through a series of large pre-trained model filters that extract object bounding boxes and masks. We then use a VLM to describe the object(s) in the bounding boxes, and use an LLM to produce several diverse object navigation labels.

paired with a learned [39, 20, 37] or classical search-based system [40, 41]. We address last-mile precision object navigation, the last portion of the L-ZSON task.

**Learning from In-the-Wild Videos:** One major bottleneck for training policies for robotics is collecting in-domain data. Several works in robot learning have studied the use of in-the-wild video data for representation and reward learning [42, 43, 44, 45, 46] or affordance and direct policy learning [47, 48, 49]. One limitation of using in-the-wild video data is the embodiment gap and lack of action labels. However, the scale and diversity of in-the-wild video data can lead to a level of policy generalization that is hard to achieve with narrow in-domain robot data.

## 3    Language-Conditioned Navigation from In-the-Wild Videos

In this paper, we study the task of "last-mile navigation" [38], where the robot must precisely navigate to an object or location that is visible from afar. Complementary to the well-studied task of semantic exploration [37, 50, 51, 10, 34], last-mile navigation focuses on fine-grained semantic understanding and precise navigation to the target object. We also show that our last-mile navigation method can be combined with a high-level planner and long-term memory to navigate in both partially seen and unseen environments.

We generate diverse language instructions and counterfactual robotic actions for action-free egocentric data. We use these instructions and actions to train a robust language-conditioned policy. In this section, we describe the LeLaN data augmentation pipeline and training methodology.

### 3.1    Labeling In-the-Wild Videos with Foundation Models

A major challenge in training language-conditioned control policies is access to high-quality and diverse robot data with language and action annotations. Moreover, teleoperated robot trajectories for general navigation often lack diversity, traveling only down the center corridors and rarely heading toward objects. Therefore, even if human-made labeling is used on existing data, obtaining rich and useful labels for object navigation remains challenging.

We propose a novel method that leverages foundation models to label in-the-wild video data for training a language-conditioned navigation policy. Fig. 2 shows the overview of our data augmentation approach. Our method is (i) scalable, as it uses abundant and readily available action-free data sources, (ii) efficient, as it leverages all captured objects in the image view during data augmentation, and (iii) generalizable, as it distills state-of-the-art large vision and language models to diverse and semantically meaningful annotations.

**Language annotation:** First, the in-the-wild video frames are passed through Segment Anything [52] to localize objects within the scene and produce masks and bounding boxes. The bounding boxes are then used to create image crops, which may include an object of interest. We then feed the image crops into an open-source visual language model (VLM) [53], which produces a description of the object or objects in the crop. This description is passed to GPT-3.5-turbo, a large language

model (LLM). If the LLM determines that the VLM confidently detected an object in the prompt, the LLM is then tasked with generating a prompt "Go to X", where X are various descriptions of the object in the image crop based on the VLM's description.

In principle one could directly use a VLM, such as GPT-4V [54], to generate multiple prompts directly from image crops. However, at the time of the submission, both the cost of the closed-source VLMs and their inference time were high, while the open-source VLMs resulted in poor quality prompts. Therefore, we leverage an open-source VLM [53] to generate image descriptions, and GPT-3.5-turbo, which is fast and inexpensive, to generate prompts from the description.

**Pose estimation and action generation:** To ground the observation in the real 3D scene, we estimate the 3D pose corresponding to the objects of interest as the goal pose and generate robot actions to move toward it without collision. For the pose estimation, we use an approach similar to CoW [34] where we approximate the depth of the scene using Metric3D [22] for the narrow FOV camera image and Depth360 [21] for the wide FOV camera image and project the result into estimated point clouds. By masking the estimated point clouds with masks from Segment Anything [52] and taking the median values, we can obtain the 3D pose corresponding to an object for which we have matching language instructions from the VLM and LLM filtering.

To learn collision-avoidant behavior, we leverage an RFM [20] for navigation to generate the robot actions with this behavior to act as supervision when training our policy. By feeding in a context of observations and the cropped goal image, we can generate the sequence of the robot actions for each object that avoids collision and moves toward the target object, from our pre-trained policy. For more details on how the RFM was used, please refer to Appendix A.

Applying our approach to the $i$-th image $o_i$ in the video data, we have $m$ objects with the corresponding estimated poses $\{p_i^j\}_{j=1...m}$ and the sequences of the robot actions $\{\bar{p}_i^j[h]\}_{h=0...M-1}$ for each object. $M$ is the control horizon of the RFM. The number of the objects $m$ is determined by the number of detections generated from Segment Anything with the GPT-3.5-turbo confidence filter. Each object has $n$ prompts as $\{l_{ig}^j\}_{g=1...n}$, where $n$ depends on the number of generated prompts.

## 3.2 Policy Architecture & Training

We train our language-conditioned navigation policy on our augmented dataset. We define the language-conditioned navigation policy as $\{v_k, \omega_k\}_{k=0...N-1} = \pi_\theta(\{o_k\}_{k=-L...0}, l)$. Our policy generates the sequence of the virtual velocity commands $\{v_k, \omega_k\}_{k=0...N-1}$ to move toward the target object pose $p_o$ by feeding the current and previous observations $\{o_k\}_{k=-L...0}$ and the prompt $l$ corresponding to the target object. Here, $v_k$ and $w_k$ are the linear and angular velocity commands respectively. $\{o_k\}_{k=-L...0}$ and $l$ are taken from our labeled dataset by randomly selecting $i$ as the current state and $j$ for the target object and $g$ for the instruction prompt. Depending on the selected $i$, $j$ and $g$, the object pose and robot action are also uniquely given, such as $p_o$ and $\{\bar{p}[k]\}_{k=0...M-1}$.

**Objective:** We define a following overall objective $J$ incorporating three objectives, $J_{\text{pose}}$, $J_{\text{col}}$ and $J_{\text{smooth}}$ to train $\pi_\theta$ encouraging object-reaching, collision avoidance and smooth trajectories:

$$\min_\theta J(\theta) := \underbrace{(\tilde{p}_o - \hat{p}_{N-1})^2}_{J_{\text{pose}}(\theta)} + \varepsilon \underbrace{\sum_{k=0}^{M-1} (\bar{p}[k] - \hat{p}_k)^2}_{J_{\text{col}}(\theta)} + \underbrace{\sum_{k=0}^{N-1} ((v_{k+1} - v_k)^2 + (\omega_{k+1} - \omega_k)^2)}_{J_{\text{smooth}}(\theta)}. \quad (1)$$

where $\{\hat{p}_k\}_{k=0...N-1} = f_{\text{robot}}(\{v_k, \omega_k\}_{k=0...N})$ is the 2D virtual robot poses from our policy, $\tilde{p}_o$ is the 2D object pose, which is the projected $p_o$ on the horizontal plane. Here, $f_{\text{robot}}(.)$ is the differentiable kinematic model. And $N(\geq M)$ is the control horizon of our policy. $J_{\text{pose}}$ is for encouraging target object reaching, $J_{\text{col}}$ is for discouraging collisions, and $J_{\text{smooth}}$ is for enabling smooth trajectories. Since our pre-trained policy learns to beeline toward the target object location without $J_{\text{col}}$, the robot can not avoid collision with objects between the robot and target objects. Hence, in addition to the base objectives $J_{\text{pose}} + J_{\text{smooth}}$, we introduce $J_{\text{col}}$ to distill the RFM, which contains an understanding of collision-avoidant behavior learned from diverse navigation datasets [20, 55]. However,

since $J_{\text{col}}$ will continue to encourage collision avoidance near target objects, we use $\varepsilon$ to mask out $J_{\text{col}}$ in the case, $|\tilde{p}_o| < 1.0$. Our training process is further described in Appendix F and G.

## 3.3 Training Data

We use a wide variety of egocentric datasets to train on, including: 1) **Indoor Navigation Dataset**: image observations from mobile robot trajectories in office building environments, 2) **YouTube Tour Dataset**: YouTube video data of tours in various countries, and 3) **Human-walking Dataset**: data collected from walking with a camera in an indoor setting and outdoor city environments. A breakdown of this data is included in Table 1. More analysis of this dataset is shown in Appendix D.

**Indoor Navigation Dataset:** The robot dataset consists of an indoor general navigation dataset [56, 57, 58] which contains autonomous and teleoperated robot trajectories, none of which explicitly contain object-reaching behavior. We employ three publicly available datasets,

Table 1: **Breakdown of the datasets annotated with LeLaN** YouTube Tour and Human-walking Dataset are sampled at 2 FPS due to the faster walking speed compared to robot speed. GS, YT, and HW indicate Go Stanford, YouTube Tour, and Human-walking, respectively.

| Datasets | | Hours [h] | Images [#] | Objects [#] | Prompts [#] |
|---|---|---|---|---|---|
| Indoor Nav. Dataset | All | 31.0 | 111570 | 594045 | 3418944 |
| | GS2 [56] | (17.6) | (63194) | (306041) | (1775073) |
| | GS4 [57] | (10.5) | (37769) | (234266) | (1338117) |
| | SACSoN [58] | (2.9) | (10607) | (53738) | (305754) |
| YT Dataset [59] | | 82.5 | 593727 | 3343956 | 14623072 |
| HW Dataset | | 15.7 | 113277 | 534544 | 2311467 |

the GO Stanford 2 (GS2) Dataset [56], the GS4 Dataset [57], and the SACSoN (HuRoN) Dataset [58] to cover a variety of office buildings and settings. Note that our data augmentation method is not limited to robot data, but includes it to increase data diversity and provide additional grounding.

**YouTube Tour Dataset:** Incorporating YouTube videos into our dataset allows us to enhance both the diversity and scalability of our approach. We curate 82.5 hours of YouTube videos that include vastly diverse domains and are significantly different from the university campus building settings seen in the Indoor Navigation Dataset. These videos cover a broad spectrum of objects, scenes, and camera heights and types, contributing to more generalized model performance. Specifically, we select videos that include home tours, outside walking (sightseeing) tour videos, and shopping mall tour videos from 32 different countries. Additionally, our dataset captures a variety of weather conditions (e.g. snowy, rainy, and sunny) and time of day (e.g. morning, afternoon, and night), and diverse environmental types, such as urban, suburban, and rural settings.

**Human-walking Dataset:** YouTube video can give us a diverse training set with many kinds of objects in various environments. However, we noticed that most of the cameras used to record YouTube videos have a narrow field of view (FOV). However, a wide FOV camera is useful for robot control [39, 57, 60]. To address this limitation, we collect a dataset of additional data by holding a wide FOV (fisheye) camera and collect a dataset by walking in inside and outside environments. The Human-walking Dataset includes 15.7 hours of data across 11 cities in 3 different countries.

## 4 Experiments

Our experiments evaluate LeLaN in the real world, studying the following research questions:

**Q1.** Can LeLaN successfully navigate to novel objects conditioned on simple and noisy or complex language instructions?

**Q2.** Can LeLaN avoid collision with obstacles between the robot and target objects?

**Q3.** Can LeLaN generalize to different embodiments?

**Q4.** How does in-the-wild video improve the learned policy performance?

To isolate the advantages from our data labeling and the collision avoidance with the RFM in evaluation, we evaluate our policy with the simplest implementation not considering $J_{\text{col}}$ as well as not feeding the history images ($L = 0$) in the evaluation. We include this objective when studying **Q2**.

Table 2: **Quantitative results in navigation using a prototype real robot.** We show the goal success rate in 150 trials with 28 objects. Success is determined by the robot reaching within a 0.2 [m] radius of the target object. YT and HW indicate the YouTube Tour Dataset and Human-walking Dataset, respectively. Inference time indicates the calculation time on Nvidia Jetson Orin to generate an action given the current observation. ∗ indicates that inference is performed only with the initial observation after which the classical controller takes over and † indicates to incorporate the classical motion planner to generate the velocity commands.

| Method | Total | Simple prompts | Noisy prompts | Multiple objects | Dynamic object | Inference time [s] |
|---|---|---|---|---|---|---|
| OpenFMNav [61]† | 0.43 | 0.38 | 0.46 | 0.31 | 0.00 | 22.0* |
| OWL-ViT [62] + ViNT [55] | 0.47 | 0.48 | 0.47 | 0.22 | 0.33 | 0.33 |
| CoW [34]† | 0.58 | 0.73 | 0.51 | 0.43 | 0.17 | 0.22 |
| OWL-v2 [63] + Zoedepth [64]† | 0.67 | 0.73 | 0.63 | 0.62 | 0.00 | 2.82 |
| Our method (YT   0.0%, HW   0.0%) | 0.65 | 0.63 | 0.66 | 0.64 | 0.67 | **0.054** |
| (YT  20.0%, HW  43.5%) | 0.85 | **0.91** | 0.82 | 0.70 | **0.83** | **0.054** |
| (YT 100.0%, HW 100.0%) | **0.89** | **0.91** | **0.88** | 0.81 | **0.83** | **0.054** |

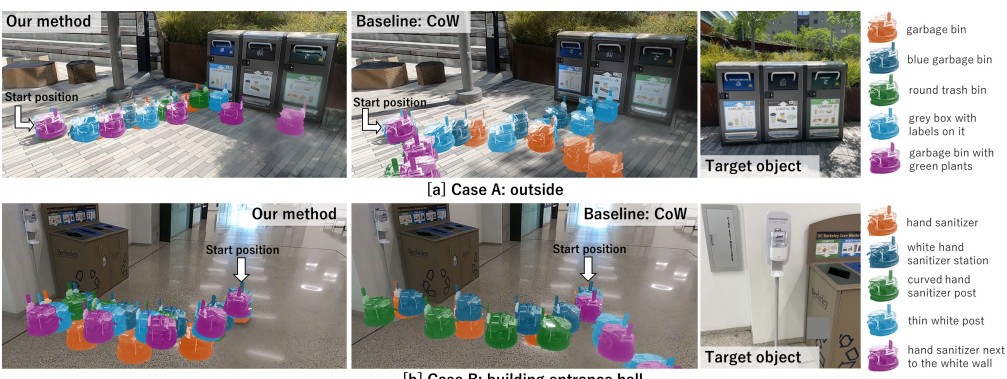

[a] Case A: outside

[b] Case B: building entrance hall

Figure 3: **Visualization of LeLaN performance.** We conduct each experiment with 5 different prompts (right side) to visualize the robustness of LeLaN against noisy prompts. Our policy can navigate the robot toward the target object along very similar trajectories, showing its performance is highly reproducible.

## 4.1   Evaluation on Diverse Language Instructions

We evaluate our trained policy on target object navigation, which tasks the robot with navigating toward a visible target object from the current robot pose. We evaluate our policy as well as four baseline methods for comparison. Please see Appendix F, G and H for details on the procedure and training hyperparameter settings, baseline methods, and a prototype robot in evaluation.

We evaluate our method on a diverse set of language instructions. We run evaluations in five different environments, including 3 indoor floors and 2 outdoor spaces, across a set of 28 objects or structures in the scene. For each of these objects, we generate a set of 5–6 language instructions of the form "Go to X". The form of X in these instructions ranges from simple, including only the object with no additional descriptors, to complex, where adjectives and relationships to other objects close to the main object in the scene are included. To evaluate the robustness of our method with a variety of instructions, we perturb the instruction in two ways: 1) by adding incorrect descriptors or similar but incorrect terms for the object and 2) by describing the object implicitly rather than explicitly. Our instructions are provided in the far right of Fig. 3 and Appendix E.

Table 2 shows the quantitative analysis from over 1050 evaluations. We categorize each instruction into two categories, simple and noisy prompts, and show the success rate for each category to demonstrate the advantage of our method. Our method achieves an average success rate of 89% compared to 67%, the performance of the strongest baseline. The model is especially robust to noisy prompts, achieving an average of 25% higher success rate on our real-world test set against the baselines. Our method still performs well on simple, correct instructions that can be challenging for the baselines. In addition, we observe that our method outperforms the baselines on prompts with multiple objects, such as "Go to the chair next to the black suitcase". More detailed discussion and analysis are provided in Appendix B and E.

Figure 3 shows a visualization of our method deployed in two scenes. In each scene, we navigate the robot towards the same object, but with 5 different prompts shown on the rightmost side. Even though we provide noisy prompts, our method successfully navigates the robot towards the same objects. However, the baseline CoW fails navigation when provided with noisy prompts.

## 4.2 Capability Analysis on Challenging Settings

To analyze the capability of LeLaN, we conduct three more challenging evaluations: 1) collision avoidance for the obstacles between the initial robot location and the target objects, 2) long-distance navigation, which the robot can not capture the target object at the initial position, and 3) following the dynamic objects such as the pedestrians.

**Collision Avoidance:** For **Q2**, we evaluate LeLaN in an environment with obstacles to determine the effectiveness of our proposed objective, $J_{col}$, for distilling collision-avoidant behavior from the RFM. We run our policies with and without $J_{col}$ as well as the strongest baseline, **OWL-v2 + Zoedepth**. We conduct 15 navigation experiments for different target objects with various obstacles and show the target reaching accuracy in Table 3.

Table 3: **Target object arrival rate in navigation with and without the obstacles.** † indicates using the classical motion planner cosidering collision avoidance.

| Method | with obstacle | wo obstacle |
|---|---|---|
| OWL-v2 [63] + Zoedepth [64]† | 0.40 | 0.67 |
| Our method wo $J_{col}$ | 0.13 | **0.89** |
| Our method with $J_{col}$ | **0.60** | 0.77 |

Since collision avoidance is a challenging task when performing language-conditioned navigation, the accuracy of our method with $J_{col}$ is 0.6. However, we can confirm that there is an explicit advantage gap compared to the best baseline method. In most cases, our method without $J_{col}$ collides with the obstacle between the robot and the target object. To add collision-avoidant behavior to the **OWL-v2 + Zoedepth** baseline, we implement classical motion planning in Appendix K. This baseline often fails navigation due to its slow inference speed. After passing the obstacles without collision, this baseline can not change the trajectory toward the target objects. The robot behaviors are shown in the supplemental video.

**Long-distance Navigation:** The LeLaN is designed for last-mile navigation, which assumes that the target object is seen at the initial robot position. However, we can extend the LeLaN to long-distance navigation by leveraging topological memory [65, 55, 66, 57]. We select the node image including the target object by scoring each node with the OWL-v2 [63] and navigate the robot toward its node image location. Then we switch the policy to the LeLaN to move toward the exact target object location. The details of our implementation are shown in Appendix J. We visualize the robot behaviors in the supplemental materials.

**Dynamic Object:** We evaluate our method and the baselines on object navigation with six dynamic objects. Here "dynamic" entails moving the object after the robot starts navigating. Due to its robust performance and fast inference speed, our method significantly outperforms the baselines as shown in Table 2. The relatively small network size allows us to implement our policy at a higher frame rate onboard (4 times faster than the fastest baseline, CoW), which is a significant advantage over the baselines that attempt to run larger models on edge compute.

## 4.3 Cross Embodiment Analysis

Given that our proposed method leverages in-the-wild videos recorded by a variety of cameras at different poses for training, it is inherently capable of generalizing to different embodiments. To rigorously evaluate our policies' cross-embodiment capabilities for **Q3**, we test our policy on the four different robot setups as shown in Fig. 4, [a] a Unitree Go1 quadruped robot with a different fisheye camera, [b] the same wheeled mobile robot as used in our extensive quantitative evaluation but with a different fisheye camera, [c] the same wheeled mobile robot with a narrow FOV camera, and [d] the same wheeled mobile robot with a spherical camera at higher height. We find that our policy can robustly navigate all of these robots toward target objects, demonstrating its ability to transfer across novel morphologies and cameras. We show each examples of the robot performance in the supplemental materials.

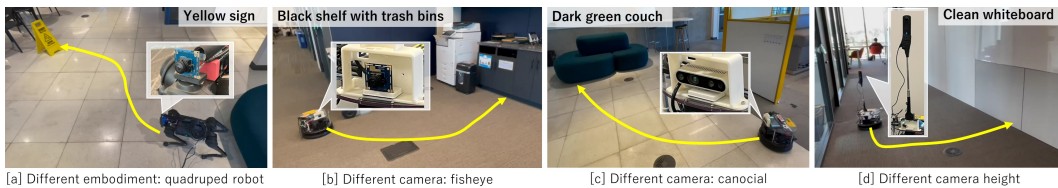

[a] Different embodiment: quadruped robot   [b] Different camera: fisheye   [c] Different camera: canocial   [d] Different camera height

Figure 4: **Overview of cross embodiment evaluation.** We conduct three type experiments to evaluate the generalized performance of our policy, [a] quadruped robot with PCB-mounted fisheye camera, [b] PCB-mounted fisheye camera, [c] canonical camera, and [d] spherical camera at higher pose

## 4.4 Data Ablations

For **Q4**, we conduct a dataset ablation study for our YouTube Tour Dataset, our Human-walking Dataset and the Indoor Navigation Dataset. We report the mean square pose error between the target object and the generated trajectories on the test dataset in Fig. 5. We start with a base mixture of 100% of the Indoor Navigation Dataset, 20% of the YouTube Tour Dataset, and 43% of the Human-walking Dataset. We train the policy with different ablations of this mixture to analyze the impact of increased data volume from each dataset.

By including more data sources in our augmented dataset, the performance of the trained language-conditioned navigation policy improves. However, performance improvements saturate with the Indoor Navigation Dataset and the Human-walking Dataset around 100K frames. The YouTube Tour Dataset significantly improves the performance beyond 100K frames and enables us to train the best policy with over 500K frames. The broad distribution of environments and objects captured in the YouTube data is the key reason that the policy's performance continues to improve. Table 2 shows an additional data ablation on navigation in the real world using our prototype robot. When increasing the proportion of these in-the-wild datasets, our method improves on all test cases.

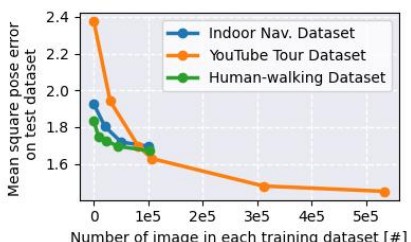

Figure 5: **Data Ablation.** An ablation of each dataset included in training data mixture, while keeping the entirety of the other datasets in the data mixture.

## 5 Conclusion

We present LeLaN, a novel approach that consumes unlabeled, action-free egocentric data and leverages foundation models to learn scalable, language-conditioned object navigation. Our work aims to take advantage of the internet-scale knowledge of foundation models such as monocular depth estimation models, VLMs, and LLMs to add language and action annotations to actionless egocentric mobile robot and human navigation data. We release a dataset of over 100 hours of in-the-wild video data, which was automatically augmented using LeLaN. Our policies can effectively scale to diverse data sources due to the generality of LeLaN. We show the importance of training our policies with in-domain robotics data. However, we also show our policies improve by including out-of-distribution from sources such as FPV walking video footage and YouTube house tours. Our policies outperform the SoTA baselines regarding precise line-of-sight object navigation with language commands of varying quality and granularity.

While our empirical results show that our policy is robust to varying environments, objects, instructions, camera types and embodiments, there are still a number of limitations that can be addressed in future work. For instance, our policy has trouble distinguishing between multiple copies of the same object using spatial reasoning alone. If there were two doors in the scene, and the instruction asked for the robot to "go to the leftmost door," the policy often gets confused among the doors. Additionally, collision avoidance remains a challenging task when training language-conditioned navigation policies. Learning to accurately distinguish target objects and understanding 3D geometry for planning will continue to be explored in future work.

**Acknowledgments**

This research was supported by Berkeley AI Research at the University of California, Berkeley and Toyota Motor North America. And, this work was partially supported by ARL DCIST CRA W911NF-17-2-0181, NSF IIS-2246811, and NSF IIS-2150826. We thank Dhruv Shah for advising the implementation for the long-distance navigation using the topological memory and the objective design for collision avoidance and Pranav Atreya for discussing the LLMs and VLMs filtering to have accurate labeling.

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

# Appendix

## A  Collision Avoidance with Supervision from Robotic Foundation Model

In our target task, last-mile navigation, we assume that there are no obstacles between the robot location and the target objects. However, we may collide with the objects that are adjacent to the initial robot location as our policy is trained to move toward the target object as directly as possible. To address this issue, we introduce the following additional objective to distill a robotic foundation model with collision avoidance capabilities into our policy.

$$J_{\text{col}}(\theta) = \sum_{k=0}^{M-1} (\bar{p}[k] - \hat{p}_k)^2,$$ (2)

where $\{\bar{p}[k]\}_{k=0...M-1}$ is the generated trajectory from the robotic foundation model such as $\{\bar{p}[k]\}_{k=0...M-1} = f_{\text{NoMaD}}(\{o_i\}_{i=-L...0}, o_g)$. Here, $M$ and $L$ indicate the control horizon of the foundation model and the length of the image history, respectively. Specifically, we employ the No-MaD [20] as the robotic foundation model, $f_{\text{NoMaD}}()$. However, as we do not assume access to the goal image $o_g$ of the target object in the context of the language conditioned object navigation task, we crop the target object from the current observation $o_0$ and feed in the cropped image as $o_g$ to generate $\{\bar{p}[k]\}_{k=0...M-1}$ in training. To generate reasonable trajectories with the cropped images, we also fine-tune the original NoMaD policy before employing it for this loss, as explained later.

To apply $J_{\text{col}}$ into our original objective $J$ and properly train our policy, we use the following techniques (Note that following techniques are only for considering $J_{\text{col}}$ in $J$.)

1. Pre-training the policy without $J_{\text{col}}$ and then fine-tuning this policy with $J_{\text{col}}$,
2. Masking out $J_{\text{col}}$ in the case that the target object is close to the robot, and
3. Feeding in the history of observations $\{o_i\}_{i=-L...0}$.

The first technique allows us to learn two competing capabilities, namely collision avoidance and goal-reaching. By first learning a goal reaching behavior, fine-tuning for an obstacle avoidance behavior is possible without compromising goal-reaching performance. For the second technique, we mask out $J_{col}$ when the target object is close to the robot location. As the robot approaches the object, the NoMaD policy will more strongly push the robot to avoid the object to avoid collision, sometimes preventing the object from being reached. The third technique, including an image history, improves the temporal consistency of the policy and performance in real experiments. We therefore follow the implementation in ViNT [55] and modify the network structure.

**Fine-tuning NoMaD policy for the cropped image:** To fine-tune NoMaD policy for image crops, we provide a crop from the current image as the goal for fifty percent of the batch and otherwise provide the original goal image. The dataset used to train NoMaD includes the 2D pose of the robot at the goal. As a result, we can randomly select a height and back-project this 3D robot frame point into an image coordinate as $[u, v]$, using this as our reference point for determining an image crop that corresponds to the target pose. We crop a randomly sized boundary box centered on this point from the current observation to use as the goal image. The rest of the training procedure remains the same. Please see the details in their original paper [20].

## B  Breakdown of quantitative experiments

We provide a break down of our quantitative experiments to analyze our results with more granularity. We categorize all environments into "Office floor", "Entrance hall", "Home environment" and "Outside" and show the mean of the target object arrival rate for each categories in Table 4. Our method trained on the full dataset outperforms all baselines in almost all categories of environments. The results clearly show the advantage of our data augmentation approach, which enables

Table 4: **Quantitative results breakdown.** We categorize all experiments into four categories, "Office floor", "Entrance hall", "Home environment" and "Outside" and show the goal success rate. Success is determined by the robot reaching within a 0.2 [m] radius of the target object. YT and HW indicate the YouTube Tour Dataset and Human-walking Dataset, respectively. † indicates to incorporate the classical motion planner.

| | Method | Total | Simple prompts | Noisy prompts | Multiple objects |
|---|---|---|---|---|---|
| **Office floor** | OpenFMNav [61]† | 0.23 | 0.25 | 0.22 | 0.17 |
| | OWL-ViT [62] + ViNT [55] | 0.35 | 0.38 | 0.34 | 0.17 |
| | CoW [34]† | 0.56 | 0.69 | 0.47 | 0.33 |
| | OWL-v2 [63] + Zoedepth [67]† | 0.65 | 0.63 | 0.66 | 0.50 |
| | Our method (YT 0.0%, HW 0.0%) | 0.77 | 0.81 | 0.75 | 0.67 |
| | (YT 20.0%, HW 43.5%) | 0.90 | **1.00** | 0.84 | 0.83 |
| | (YT 100.0%, HW 100.0%) | **0.96** | **1.00** | **0.94** | **1.00** |
| **Entrance hall** | OpenFMNav [61]† | 0.42 | 0.50 | 0.38 | 0.33 |
| | OWL-ViT [62] + ViNT [55] | 0.35 | 0.40 | 0.31 | 0.17 |
| | CoW [34]† | 0.50 | 0.70 | 0.38 | 0.50 |
| | OWL-v2 [63] + Zoedepth [67]† | 0.57 | 0.70 | 0.50 | **0.83** |
| | Our method (YT 0.0%, HW 0.0%) | 0.50 | 0.40 | 0.56 | 0.33 |
| | (YT 20.0%, HW 43.5%) | **0.85** | **0.90** | **0.81** | **0.83** |
| | (YT 100.0%, HW 100.0%) | **0.85** | **0.90** | **0.81** | **0.83** |
| **Home env.** | OpenFMNav [61]† | 0.60 | 0.58 | 0.61 | 0.0 |
| | OWL-ViT [62] + ViNT [55] | 0.23 | 0.42 | 0.11 | 0.10 |
| | CoW [34]† | 0.60 | 0.67 | 0.56 | 0.40 |
| | OWL-v2 [63] + Zoedepth [67]† | 0.77 | **0.83** | 0.72 | **0.80** |
| | Our method (YT 0.0%, HW 0.0%) | 0.53 | 0.42 | 0.61 | 0.70 |
| | (YT 20.0%, HW 43.5%) | 0.83 | **0.83** | 0.83 | 0.70 |
| | (YT 100.0%, HW 100.0%) | **0.87** | **0.83** | **0.89** | **0.80** |
| **Outside** | OpenFMNav [61]† | 0.52 | 0.67 | 0.43 | 0.70 |
| | OWL-ViT [62] + ViNT [55] | 0.59 | 0.56 | 0.61 | 0.30 |
| | CoW [34]† | 0.65 | 0.78 | 0.57 | 0.40 |
| | OWL-v2 [63] + Zoedepth [67]† | 0.67 | 0.78 | 0.61 | 0.60 |
| | Our method (YT 0.0%, HW 0.0%) | 0.65 | 0.72 | 0.61 | **0.80** |
| | (YT 20.0%, HW 43.5%) | 0.80 | 0.83 | 0.79 | 0.40 |
| | (YT 100.0%, HW 100.0%) | **0.87** | **0.89** | **0.86** | 0.70 |

us to leverage the in-the-wild videos to learn a generalized policy. It is evident that even with just the Indoor Navigation dataset, which does not include at-home and outdoor data, our method still exhibits a clear performance gap between the baselines. However, the gap narrows between our method and the baseline OWL-v2 + Zoedepth because it is strong in home environments. The baseline methods are relatively weak in the office floor and entrance hall environments, which are more semantically sparse compared to the home environment. This weakness leads to a large gap in the mean score between our method and the baselines across all evaluations.

## C  Model Ablations

We also study how ablations of our model architecture impact the performance of our policy while training on the LeLaN dataset. For the visual encoder, we replace "ResNet-FiLM" in our method with "ViT-B32" and "ViT-ResNet50" of the pre-trained CLIP. For the text encoder, we employ a larger pre-trained CLIP text encoder "ViT-ResNet50" instead of "ViT-B32".

In this ablation study, we evaluate each model on the test dataset. Similar to the objective in training, we calculate the mean square error between the generated virtual robot pose from the control policy and the target object pose. From Table 5, the pre-trained visual encoders from CLIP are worse than the ResNet-FiLM trained on our augmented dataset. The visual features from the CLIP visual encoder are insufficient to derive time-series velocity commands because they do not include geometric information. Furthermore, the ResNet-FiLM inserts the text features from the text encoder for low-level visual features, which helps to understand the target objects in the image view. In addition, the larger CLIP text encoder helps with learning a precise control policy. However, the advantage of a larger encoder is not significant on the test dataset. Furthermore, when navigating with a real robot, its difference was trivial and, in fact, increased the computational load on the robot

controller. This not only reduced the frame rate, but also increased battery consumption. Therefore, we used the the pre-trained text encoder from the "ViT-B32" CLIP model for the main model in the paper.

Table 5: **Ablation study of our model architecture.** We use the pre-trained weights of ViT-B32 and ViT-ResNet50 from CLIP for both the visual and text encoders. When using ResNet-FiLM, we train our model from scratch.

| Visual encoder | | | Text encoder | | MSE |
|---|---|---|---|---|---|
| ResNet-FiLM | ViT-B32 | ViT-ResNet50 | ViT-B32 | ViT-ResNet50 | |
| ✓ | | | ✓ | | 1.291 |
| ✓ | | | | ✓ | 1.202 |
| | ✓ | | ✓ | | 1.690 |
| | | ✓ | | ✓ | 1.673 |

# D   Frequency of the object noun in our augmented dataset

We visualize the frequency of object nouns in our augmented dataset in Fig 6. We can confirm that our dataset includes a wide variety of nouns allowing us to train a generalized policy, which can navigate towards various objects. The YouTube Tour Dataset effectively augments the category distribution with objects from home environments and outdoors, for which the Indoor Navigation Dataset is not sufficient.

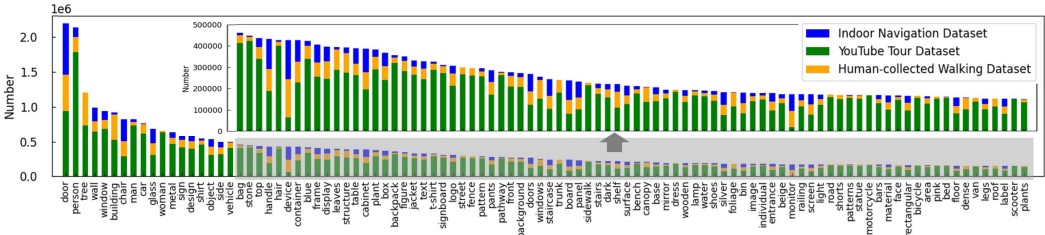

Figure 6: **Top 100 nouns in generated instructions.** Blue, green, and orange bars indicate the Indoor Navigation Dataset, YouTube Tour Dataset, and Human-walking Dataset, respectively.

# E   Target Objects and Prompts in Evaluation

In our evaluation, we select 18 objects in the university campus environment (inside and outside) and 10 additional objects in the home environment and prompt the robot to navigate towards the target objects. For each object, we feed 5 or 6 trials with different prompts (some of which are noisy) and evaluate the robustness of the policy. Here, we show the overview of the target objects and the prompts in our evaluation. The first 18 objects are from the university campus and the rest are from a home environment.

First, two prompts for each object are for the simple prompts and the others are for noisy prompts, which includes wrong adjectives (red) long prompts, or the prompts without the target object's noun. The red border in the image indicates the presence of multiple corresponding objects in the experimental environment. If the objects can be distinguished by prompts, success is considered only if the robot reaches the correct object; if the objects cannot be distinguished by prompts, success is considered if the robot reaches one of the objects that fits the description of the prompt.

- couch
- small rounded purple couch
- purple carpet
- round plush piece of furniture
- purple couch on the grey concrete floor, next to the blue grey pillow
- pinkish purple cushion-like stool

- fridges
- shiny fridge with a water dispenser
- arched fridge with a water server
- tall shiny appliance with two doors
- Fridge next to the wooden shelf
- dark silver something to storage food

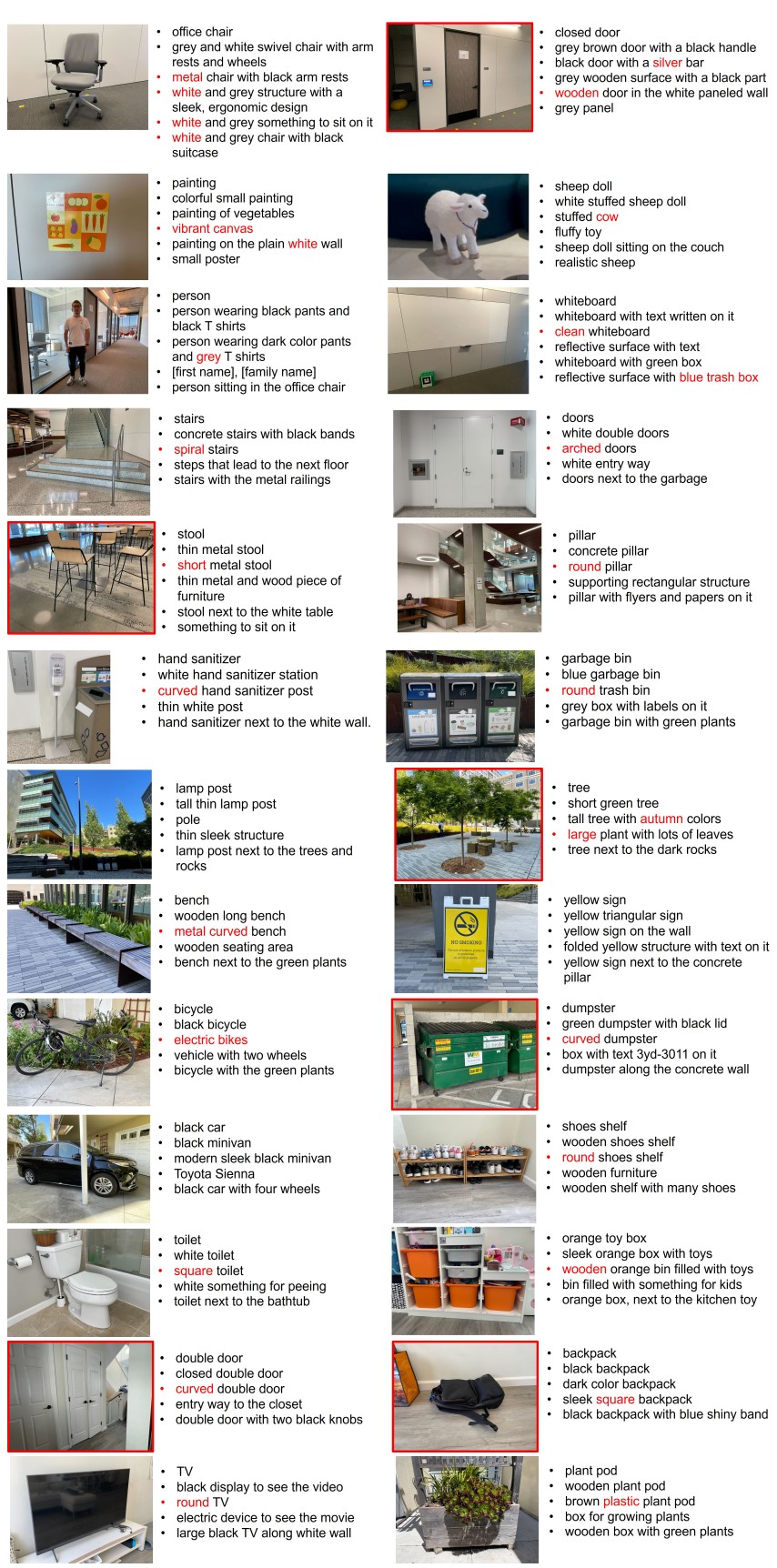

Figure 7: **Overview of 28 target objects and various prompts in our evaluation.**

## F Network architecture:

Figure 8 shows the overview of our network architecture, $\pi_\theta$. We encode the prompt $l$ with a frozen CLIP text encoder. Using a visual encoder with ResNet-FiLM [68], we have the visual feature conditioned on the prompt $l$. In addition, similar to [55, 20], we take the visual feature of $\{o_k\}_{k=-L\ldots0}$ with EfficientNet-B0 and feed these extracted features to a set of Transformer and fully connected MLP layers to produce the sequence of linear and angular velocities $\{v_k, \omega_k\}_{k=1\ldots N}$. The ResNet-FiLM encoder allows the model to learn low-level and high-level between the image observation and the text embedding. This helps the model generate the proper velocity commands to move toward the target object pose corresponding to the prompt $l$.

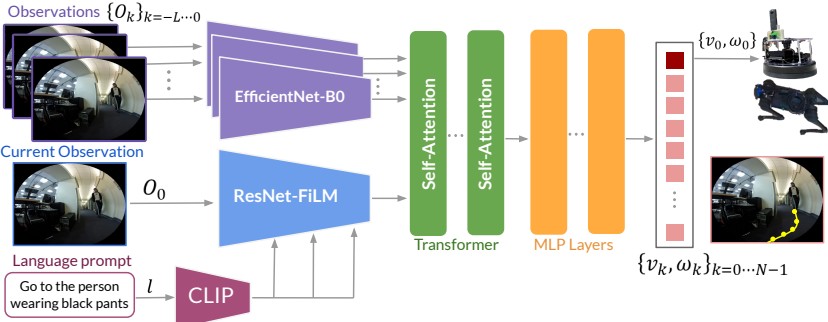

Figure 8: **Architecture.** The observations and language prompt are passed into the model. The prompt is encoded by the CLIP text encoder and passed to the ResNet-FiLM which also takes in the current observation. TheThe embedding is then passed to an MLP which produces a sequence of 24 linear angular velocity pairs which can be integrated into a trajectory.

## G Implementation details

We show the details of our training and the evaluation setup using a real prototype robot for language-conditioned navigation.

### G.1 Training

To train our control policy, we randomly choose 256 observations from our whole dataset. Since one observation contains multiple objects and each object contains multiple prompts, in almost all cases, we randomly select the object and prompt (which is based on the object). We select $L = 1$ to feed only the previous observation (1 second before in YouTube Tour Dataset) to learn the temporal consistency.

By feeding the observations and the prompts into the model, we calculate our model and generate a sequence of the velocity commands for $N(=24)$ steps. Then, we estimate the virtual robot pose $N$ steps in the future via our kinematic model (integration of the velocity commands in our case). Finally, we calculate the objective $J$ in Eqn. 2 and update our policy $\pi_\theta$. We freeze the CLIP text encoder ("CLIP" in Fig. 8) to keep the original performance during training. Our training is with an Adam optimizer using a learning rate 0.0001 on a workstation with a Intel i9 CPU, 96GB RAM, and an NVIDIA RTX 4090 GPU.

Similar to the other machine learning works with multiple objectives, we required some trial and error to find good weighting between the goal-reaching and collision avoidance terms of our objective. For example, a much larger weight for $J_{\text{col}}$ causes degradation of the goal reaching performance, which should be learned via $J_{\text{pose}}$. Moreover, we mask out $J_{\text{col}}$ by setting $\eta$ to 0 or 1 $|p_o|$ is less than a certain distance threshold. Since the actions from the RFM try to avoid collisions, even with target objects, $\eta$ prevents this behavior from dominating close to the goal. Although our approach does show an explicit advantage against the baselines, the target object reaching rate is not

sufficient for the practical use. Hence, we still believe that reliable collision avoidance with precise language-conditioned navigation is a remaining challenge for our method.

## G.2   Robot experiments

Figure 9 shows the overview of a prototype mobile robot in navigation. We calculate the control policy on the edge robot controller, Nvidia Orin AGX, with the best frame rate for each method. We mount the omnidirectional camera, a RICOH Theta S on the robot and only use the front-side fisheye camera as the observation in our base evaluation. Since we learn the visual encoder from scratch, there are no restrictions on the camera on the robot. In the cross embodiment analysis, we replace a RICOH Theta S with a PCB-mounted fisheye camera and a narrow FOV camera (Intel D435i) and additionally change the camera height. Our policy works well across these various camera configurations, but a wide FOV camera is generally better for reducing blind spots and making object detection easier.

In the evaluation, we provide the language instruction to the policy once at the beginning of navigation to reduce the computational load in each step, calculating the text encoding only once with the CLIP encoder. To control the real robot, we repeatedly calculate our control policy at the best frame rate on the robot edge controller and feed the first step velocity command $\{v_0, \omega_0\}$ in the generated sequence of the velocity commands $\{v_k, \omega_k\}_{k=0...N}$, similar to the receding horizon control.

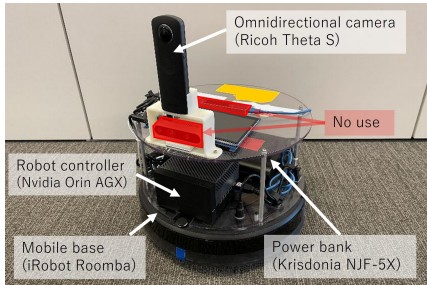

Figure 9: **Overview of the prototype mobile robot.** Note that we only use the front side camera of the omnidirectional camera, Ricoh Theta S, to navigate the robot. And we replace the Ricoh Theta S with various camera types in the cross embodiment analysis.

# H   Baseline Methods

In our evaluation, we conduct a comparative evaluation with four strong baselines trained on the internet scale datasets. In addition, we perform evaluations with policies trained on ablated data mixtures, which we discuss in the main paper. Here we explain the details of each of the four baseline implementations.

**CLIP on Wheels (CoW) [34]**   We implement the best-performing CoW baseline with the OWL-ViT B/32 detector [62]. Similar to our method, we feed the current observation and the prompts corresponding to the target object into the OWL-ViT B/32 detector [62], which was trained an internet-scale dataset to estimate the boundary box for the target object. We crop the estimated point clouds by the estimated boundary box and take the median value as the target object pose. To have a fair comparison with our method, which uses a single camera image, (no depth camera and LiDAR), we estimate the depth with Depth360 [21] and project it to estimate the point clouds. To control the robot toward the detected object, we use a state lattice motion planner to generate the linear and angular velocity commands. The details of the implemented state lattice motion planning are shown below in this appendix. We limit the scope of instructions to object navigation for objects within view from the starting point of the robot trajectory. Therefore, we do not implement the exploration portion of CoW [34].

**OWL-v2 [63] + Zoedepth [67]**   In addition to the CoW implementation, we implement a version with more performant models, including OWL-v2 as the fine-tuned VLM and Zoedepth [67] for monocular depth estimation to detect the target object pose from the current observation and the prompt, instead of the OWL-ViT and the Depth360 [21] in CoW [34]. To move toward the detected object locations, we use the same state lattice motion planner as with our CoW implementation to generate linear and angular velocity commands. Due to the larger patch size in OWL-v2, the inference speed is lower than the CoW.

**OWL-ViT [62] + ViNT [55]**   To compare our method with a learning-based method, we leverage the foundation model for vision-based navigation, which can navigate the robot towards a goal position conditioned on a goal image view. To take a goal image view corresponding a target object, this baseline leverages OWL-ViT, a VLM trained on internet-scale data, and combine it with ViNT [55], a control policy trained on multiple dataset collected by various mobile robots. Specifically, we feed the cropped image from the bounding boxes from OWL-ViT into ViNT as the goal image.

**OpenFMNav [61]**   We implement another zero-shot method for object navigation that targets robust performance with noisy language instructions. While this method also implements a method to explore semantic frontiers to locate goal objects, we simplify the task to our scope, meaning that we modify the method for last-mile navigation. With this modified version, the language instruction is fed into a VLM, in this case GPT-4o, to generate a list of potential objects to look for in the scene. Then, the observation from the robot is fed into a VLM to identify objects in the scene, including objects that may be or are semantically similar to the target objects. We use the exact same prompts as used in the original OpenFMNav implementation. These sets of objects are grouped and fed into Grounded-SAM to get potential object segmentations. We use the same depth estimation model, Depth360 [21], and the same state lattice motion planner as in the CoW [34] baseline. We implemented two versions of this method, one where inference was performed on each observation and another where the state lattice planner takes over after the object is located. As the first implementation's inference speed was extremely slow and less performant then the second option, we use the second in our baseline evaluations.

# I   YouTube Video List

We list all URLs of the YouTube video in our YouTube Tour Dataset below.

- https://www.youtube.com/watch?v=vQU_QydOUIw
- https://www.youtube.com/watch?v=5J2Wsvnk-Ec
- https://www.youtube.com/watch?v=b9thcSOI8bw
- https://www.youtube.com/watch?v=V511PNMx2uw
- https://www.youtube.com/watch?v=DO-JDTu_h5I
- https://www.youtube.com/watch?v=oQ61ijCHego
- https://www.youtube.com/watch?v=E0Jiu-1cx40
- https://www.youtube.com/watch?v=EwJQG74bl74
- https://www.youtube.com/watch?v=rMOlDH0bv1o
- https://www.youtube.com/watch?v=9MWWZeCr3QE
- https://www.youtube.com/watch?v=-3vt2Mylvsw
- https://www.youtube.com/watch?v=HknDp84cFBM
- https://www.youtube.com/watch?v=l_s9YAluXBY
- https://www.youtube.com/watch?v=k3Q1vse7In8
- https://www.youtube.com/watch?v=ISNDJ2Pjq34
- https://www.youtube.com/watch?v=4jDa_5S-0W4
- https://www.youtube.com/watch?v=2O7JrGu_mVk
- https://www.youtube.com/watch?v=je8267s9z38
- https://www.youtube.com/watch?v=tWovplr-ois
- https://www.youtube.com/watch?v=UmCbkpRUOA4

- https://www.youtube.com/watch?v=Ea2yExKlg7w
- https://www.youtube.com/watch?v=1Zu6Xct5bLQ
- https://www.youtube.com/watch?v=9IluzedLtYs
- https://www.youtube.com/watch?v=lnYfw_ryOdQ
- https://www.youtube.com/watch?v=9r5eK5JXzLo
- https://www.youtube.com/watch?v=LdWHy-f3jYg
- https://www.youtube.com/watch?v=Kcc7zuQDlpE
- https://www.youtube.com/watch?v=r-98ADAXxQM
- https://www.youtube.com/watch?v=iRfQa2SEu0Q
- https://www.youtube.com/watch?v=NzFbFARYhfE
- https://www.youtube.com/watch?v=Eo0lY4s85hE
- https://www.youtube.com/watch?v=stUYODYcPCI
- https://www.youtube.com/watch?v=GCkYfI5LRYM
- https://www.youtube.com/watch?v=848EpwPmQfA
- https://www.youtube.com/watch?v=Bq4rmeIvJbs
- https://www.youtube.com/watch?v=Fbtoj1dT8l4
- https://www.youtube.com/watch?v=QVh0TuWyEYc
- https://www.youtube.com/watch?v=2h3t7nZG2tY
- https://www.youtube.com/watch?v=hlOLZNZjb9E
- https://www.youtube.com/watch?v=ZrTNUygNt6g
- https://www.youtube.com/watch?v=BSxV9nUfDAU
- https://www.youtube.com/watch?v=c76zh6FppMQ
- https://www.youtube.com/watch?v=BP0aZVaUPF4
- https://www.youtube.com/watch?v=teGXoYsmcT0
- https://www.youtube.com/watch?v=dUtIcmgefa4
- https://www.youtube.com/watch?v=OjHbS-_nncw
- https://www.youtube.com/watch?v=q1Y9pQNcx6I
- https://www.youtube.com/watch?v=ob7F3nUFzyU
- https://www.youtube.com/watch?v=8ynE2JA8kQE
- https://www.youtube.com/watch?v=1ijW90dI9IQ
- https://www.youtube.com/watch?v=oz1Mgu8e1N4
- https://www.youtube.com/watch?v=rZwXrjFFWcw
- https://www.youtube.com/watch?v=cxgx-2iipXo
- https://www.youtube.com/watch?v=jzLnLeA-JOc
- https://www.youtube.com/watch?v=_OZhGsKdBYY
- https://www.youtube.com/watch?v=MUfHc7cIxV0
- https://www.youtube.com/watch?v=NRah8cgOB7s
- https://www.youtube.com/watch?v=PGvkZOfp-Cw
- https://www.youtube.com/watch?v=cv4WJBKJWsE
- https://www.youtube.com/watch?v=9d8szg8LqV0
- https://www.youtube.com/watch?v=Uu6PHveo_TQ
- https://www.youtube.com/watch?v=or0YX800OJ0
- https://www.youtube.com/watch?v=XVUiJH4SSxI
- https://www.youtube.com/watch?v=pQ-ogEzwu5U
- https://www.youtube.com/watch?v=3CRJWy5zXvg
- https://www.youtube.com/watch?v=5ieD4hXK7ng
- https://www.youtube.com/watch?v=Y1AEjBtkF-M
- https://www.youtube.com/watch?v=epDp5YVYpNc
- https://www.youtube.com/watch?v=o4fkFMTpKVE
- https://www.youtube.com/watch?v=JB0A8Me8EKk
- https://www.youtube.com/watch?v=7UmPa6BIjGg
- https://www.youtube.com/watch?v=N787WRdI35A
- https://www.youtube.com/watch?v=uIuWlkP6RIo
- https://www.youtube.com/watch?v=NLbp09bIxbc

- https://www.youtube.com/watch?v=yJ--4YVsRuQ
- https://www.youtube.com/watch?v=MMyPtP0wHzA
- https://www.youtube.com/watch?v=k7V3aerwT0k
- https://www.youtube.com/watch?v=0WFAbXjpC1s
- https://www.youtube.com/watch?v=TD6rh0LtWQI
- https://www.youtube.com/watch?v=6T1FiFmNKws
- https://www.youtube.com/watch?v=atud53sOPSg
- https://www.youtube.com/watch?v=bpE94DqS-v0
- https://www.youtube.com/watch?v=nXMjc6f1CTY
- https://www.youtube.com/watch?v=DgSId18eRsM
- https://www.youtube.com/watch?v=57c8dLesC-8
- https://www.youtube.com/watch?v=kxzPTywmXkc
- https://www.youtube.com/watch?v=zPa0k3yMy5Y
- https://www.youtube.com/watch?v=YCsJK3ze2_c
- https://www.youtube.com/watch?v=D3yqBIZULh0
- https://www.youtube.com/watch?v=G4o48y49S-c
- https://www.youtube.com/watch?v=dv2rrT7He9A
- https://www.youtube.com/watch?v=IVzkvNr0vHg
- https://www.youtube.com/watch?v=9FOhMsPOfhM
- https://www.youtube.com/watch?v=HryNemlCnHU
- https://www.youtube.com/watch?v=wQ9Jw0nq2Zc
- https://www.youtube.com/watch?v=iJb8P2fD15E
- https://www.youtube.com/watch?v=zuQ2KRZdCcM
- https://www.youtube.com/watch?v=J6A5rJT0yQI
- https://www.youtube.com/watch?v=1Wd6cJtOdHk
- https://www.youtube.com/watch?v=BZtiTxWl9OI
- https://www.youtube.com/watch?v=yi5bdpZb5mI
- https://www.youtube.com/watch?v=I2qb9Bf2gsw
- https://www.youtube.com/watch?v=Sar9PS_S_Kw
- https://www.youtube.com/watch?v=s780sEOPrQ4
- https://www.youtube.com/watch?v=SiOohX9vKnA
- https://www.youtube.com/watch?v=jM45nECw_hE
- https://www.youtube.com/watch?v=tsR_G4as2fQ
- https://www.youtube.com/watch?v=C_jSIKC1OyY
- https://www.youtube.com/watch?v=e_95v3g7P78
- https://www.youtube.com/watch?v=gI2waPs5wKQ
- https://www.youtube.com/watch?v=KBJHDSUkXSs
- https://www.youtube.com/watch?v=VAz1mX7F4vk
- https://www.youtube.com/watch?v=eJ7XDfP1O-U
- https://www.youtube.com/watch?v=sMqBhBCE9ac
- https://www.youtube.com/watch?v=PS2Ed9NTmbw

## J  Long-distance navigation

Before running LeLaN with topological memory for long-distance navigation, we collect the topo-
logical memory (a sequence of images) by teleoperating through the environment. To navigate
toward the target object corresponding to the prompt, we search all node images with OWL-v2 [63].
By feeding the node image and the prompt into Owl-v2, we are provided a score that corresponds
to the probability of the object's existence in each node image. Then, we select the node ID number
with the highest score and navigate the robot toward its node ID with the vision-based navigation
system [65, 55, 66, 57]. In our implementation, we employ ViNT as the vision-based navigation
system.

When arriving at the selected node, we switch the policy from ViNT to LeLaN and control our robot toward the target object position. Note that the position of the selected node ID is generally not adjacent to the target object. We utilize ViNT's temporal distance estimation to switch the policy. Specifically, when the closest node is the selected node, we switch the policy.

# K  State lattice motion planning

We implemented sampling-based motion planning as the local motion planner in **CLIP on Wheels (CoW)**, **OpenFMNav** and **OWL-v2 + Zoedepth** baselines. We generated 15 motion primitives assuming steady linear and angular velocity commands for 8 steps (2.664 s). The pairs of linear and angular velocity commands are $(v_s, \omega_s)$ = (0.0, 0.0), (0.2, 0.0), (0.2, 0.3), (0.2, 0.6), (0.2, 0.9), (0.2, $-0.3$), (0.2, $-0.6$), (0.2, $-0.9$), (0.5, 0.0), (0.5, 0.3), (0.5, 0.6), (0.5, 0.9), (0.5, $-0.3$), (0.5, $-0.6$), (0.5, $-0.9$). We selected these 15 motion primitives by balancing computational load and navigation performance.

By integrating these velocity commands for 8 steps, we obtained 15 trajectories such as $\{\{^s p_i^j\}_{i=1\ldots8}\}_{j=1\ldots15}$, where $^s p_i^j$ is the $i$-the virtual robot pose on the $j$-th motion primitive. To select the best motion primitive, we calculated the following cost value for each primitive.

$$J_s^j = \min_i (\tilde{p}_o - {}^s p_i^j)^2 + C_{ob} \tag{3}$$

Here, $\tilde{p}_o$ indicates the estimated target object pose in each baseline. The first term calculates the squared errors between all 8 poses in the $j$-th motion primitive and the goal pose to consider the goal reaching. The second term $C_{ob}$ is a constant value used to filter out trajectories that collide with static obstacles. We calculate $C_{ob}$ as follows:

$$C_{ob} = \begin{cases} 1000.0 & (\text{if } d_s < r_r) \\ 0.0 & (\text{otherwise}) \end{cases}, \tag{4}$$

where $d_s$ is the minimum distance between all 8 poses in the $j$-th motion primitive and the estimated point clouds corresponding to the static obstacles. Similar to SACSoN [58] and ExAug [66], a collision is determined when the distance between the static obstacle and the robot is less than the robot's radius $r_r$. To ensure fair evaluation with other methods that only uses an RGB camera, we utilize estimated point clouds from the current observation of the RGB camera in each method. We choose the motion primitive with the minimum $\{J_s^j\}_{j=1\ldots15}$ and assign the corresponding velocity commands $v_s$ and $\omega_s$ to control the robot during navigation.

