# OpenReview forum: "LeLaN: Learning A Language-Conditioned Navigation Policy from In-the-Wild Video"
_robot-learning.org/CoRL/2024/Conference — CoRL 2024_

### Official Review · Reviewer_W8xr · 2024-07-17
**Original Review for LeLaN: decent contribution and improvement but can target more challenging key problems**

**Originality:** 3
**Technical Quality:** 2
**Clarity Of Presentation:** 3
**Potential Impact:** 3
**Recommendation:** 3
**Confidence:** 4

**Review:**

Strengths:
- The basic idea is easy to follow, and the motivation is clearly presented.
- The overall writing is fine enough to convey the key points and ideas, except for several typos that should be polished afterward.
- The proposed pipeline is scalable for leveraging in-the-wild data, which is basically automatic.
- The experimental results show its effectiveness compared to previous works, with a significant improvement shown in the benchmark.

Weaknesses:
- There are obvious limitations regarding the data generation pipeline. For example, it does not take into account the obstacle avoidance problem in this process. In addition, from the visualization examples, the navigation target is typically in the local region, without touching more challenging long-distance navigation with more complex localization and mapping situations. Therefore, from the perspective of the overall navigation system, it does not tackle key challenges in long-distance planning or (even static) obstacle avoidance in local planning. This is a little disappointing and weakens the contributions significantly. The key problem, from my view, is the way to generate the navigation ground truth path, that can be improved in the future.
- The experiments are a little weak, including only two baselines on one main benchmark and only one ablation about the used data. I find new results in the supplementary material but may still miss some important ablations, such as the performance with different foundation models for the data generation pipeline.
- The overall paper seems to be a rushed one, considering the typos and the overall organization of materials in the main paper and supplemental. There should be a large room to be refined and I would recommend the author carefully make the revision according to the reviews during the rebuttal phase.

**Quality Of The Limitations Section:**

3

**Questions For Rebuttal:**

See weaknesses.

**Robotics Focus:**

4

**Summary Of Paper:**

This paper proposes a method, LeLaN, that uses action-free egocentric data to learn language-conditioned object navigation. Specifically, it leverages LLM and vision foundation models to label in-the-wild data for augmentation and trains a policy that can outperform state-of-the-art methods. Experiments validate the efficacy of the proposed method.

**Summary Of Recommendation:**

The paper targets the problem of leveraging in-the-wild data for a stronger navigation policy. The pipeline and generated data are validated effective via both quantitative benchmarks and hardware experiments. Although it does not tackle more challenging problems such as local planning that can avoid obstacle collision or long-distance navigation, it indeed makes progress in the corresponding niche and shows great improvement compared to previous works. Therefore, I vote for weak acceptance and recommend the author carefully consider the reviews and strengthen the submission during the rebuttal phase.

---

### Official Review · Reviewer_J4Ez · 2024-07-21
**Nice demonstrations, but the simplicity of the task and comparison to existing work could be improved**

**Originality:** 2
**Technical Quality:** 2
**Clarity Of Presentation:** 4
**Potential Impact:** 2
**Recommendation:** 3
**Confidence:** 4

**Review:**

Quality: Good. Novel approach to language-conditioned navigation. Several experiments and analysis.

Clarity: Well-structured and clearly written.

Originality: Several recent works have shown how datasets that have been pseudo-labeled using VLMs and LLMs can be used to train capable models. However, the authors apply this approach specifically to the task of ObjectNav, which is not as common.

Significance: Not sure how useful the task is. The approach is mainly for navigating towards objects that are already in front of the robot and in view. The authors recognize this and acknowledge that several existing works go farther; these other works address the case in which the target object is farther away and not in view.

Strengths:
- Use of diverse data sources, including YouTube videos and human-collected data
- Improvements over baseline methods
- Thorough experimentation and ablation studies
- Release of a large, annotated dataset for future research
- Addresses the challenge of expensive human-annotated datasets
- Demonstrates robustness to noisy and complex language instructions

Weaknesses:
- Limited comparison to only two baseline methods. The authors state that LeLaN surpasses methods that are state-of-the-art in ObjectNav (thus attaining a new state-of-the-art in ObjectNav), but I don’t think this is quite the case. There have been several works that have shown open-vocabulary ObjectNav performance in the real world, and unlike LeLaN, these methods can search for the object even when it isn’t in view (see references below). Many of these methods have shown greatly superior performance on the ObjectNav task within benchmarks such as HM3D compared to CoW. Furthermore, these methods are zero-shot, and do not require generating a bespoke dataset for ObjectNav nor any training or fine-tuning at all.
- Lack of discussion on computational requirements and scalability of the approach. What was the cost of using a closed-source LLM like GPT 3.5 to label all of the samples? How many GPUs were used to generate the dataset, and how long did it take?
- The simplicity of the policy’s architecture suggests that it will not fare well for more complex ObjectNav tasks that require temporal reasoning. The LeLaN policy only considers information that is available in the current time step.

References:
InstructNav: https://arxiv.org/abs/2406.04882
OpenFMNav: https://arxiv.org/abs/2402.10670
VLFM: https://arxiv.org/abs/2312.03275

**Quality Of The Limitations Section:**

3

**Questions For Rebuttal:**

This method seems overly-complicated, while the task seems too simple. It is not clear to me that this approach has benefits over other existing approaches that work in the real world (see references below), several of which don’t require generating a dataset or training (i.e., are zero-shot). It seems like this method could easily be replaced with a combination of a metric depth estimator and a referring expression model (e.g., ONE-PEACE, UNI-NEXT, MDETR). These models are lightweight, and can also be run onboard a robot. Once the pose of the object is computed, it is simple to navigate to its coordinates. What benefits does generating the LeLaN dataset and training on it have over an approach like this?

The paper claims to achieve state-of-the-art ObjectNav performance, but only compares against two baseline methods, one of which is CoW, which is a relatively older method that has since been outperformed by several papers that cite it. And I don’t think ViNT is a state-of-the-art approach for ObjectNav, I believe its focus was more on ImageNav. I understand that you are feeding ViNT image chips from an object detector, but ViNT was designed to take in a full image, rather than just a box of one object. Further, if you are already assuming there is only one target object that is in view, a referring expression model would be more appropriate than an object detector. Could you add experiments to compare against other relevant state-of-the-art approaches?

Please provide details on the computational resources required for dataset generation. How many GPUs were used, and what was the total time taken? This is important for assessing the scalability and reproducibility of the proposed method.

References:
InstructNav: https://arxiv.org/abs/2406.04882
OpenFMNav: https://arxiv.org/abs/2402.10670
VLFM: https://arxiv.org/abs/2312.03275

**Robotics Focus:**

4

**Summary Of Paper:**

LeLaN is an approach to language-conditioned navigation that leverages large vision-language models (VLMs) to automatically generate instructions and action labels for diverse egocentric video data. This method enables the creation of scalable, language-annotated navigation datasets from various sources, including robot observations, YouTube videos, and human-collected walking data. By utilizing the knowledge of foundation models, LeLaN aims to ground language understanding for embodied navigation. The approach demonstrates good performance in robustness to noisy and diverse language instructions, as well as precision in reaching target objects. Empirical results highlight the effectiveness of incorporating diverse data sources, including out-of-distribution data. The authors argue that this method overcomes limitations of expensive human-annotated datasets and bridges the gap between foundation model knowledge and embodied robotic experiences. The research contributes a dataset generation technique, a navigation policy, and data ablation studies that emphasize the importance of in-domain robotics data.

**Summary Of Recommendation:**

This paper presents LeLaN, a novel approach to language-conditioned navigation using large vision-language models to generate datasets from diverse video sources. While the method shows promise in handling noisy instructions and improving upon some baselines, there are concerns about its practicality and state-of-the-art claims. The limited scope of the task (navigating to visible objects) and the complexity of the data generation process may not justify the benefits over simpler, zero-shot approaches. The comparison to only two baselines, which may not represent current state-of-the-art, is a notable weakness. Additionally, the lack of information on computational requirements and limited exploration of failure cases should be addressed to understand scalability and robustness. While the paper contributes to the field, addressing these concerns and providing a more comprehensive comparison with recent methods would significantly strengthen the work. The authors should consider revising their claims and expanding their experimental comparisons to better position their contribution within the current state of object navigation research.

---

### Official Review · Reviewer_DySg · 2024-07-22

**Originality:** 2
**Technical Quality:** 3
**Clarity Of Presentation:** 4
**Potential Impact:** 3
**Recommendation:** 3
**Confidence:** 3

**Review:**

Strengths:
- Well-written. The paper is well written and easy to follow.
- Visualizations. The figures very much aid in understanding the paper.
- Experimental Results. The paper compares to and outperforms a number of strong baselines
- A number of ablations are conducted which validate the various design choices made.
- Qualitative evaluation is included

Weaknesses:
- Problem statement. The authors write: "We limit the scope of instructions to object navigation for objects within view from the starting point of the robot trajectory." Even from the visualization, in Figure 5, it looks like the robot starts quite close to the target object. This seems to greatly simplify the problem, while recent works in navigation are tackling semantic exploration real world environments.
- Weak baselines. The authors write: "The relatively small network size allows us to implement our policy at a higher frame rate onboard, which is a significant advantage over the baselines that try to run larger models on edge compute." The baselines were not necessarily designed for edge compute -- if the baselines were given the same number of attempts at inference, how would they perform? Furthermore, is this difference only for dynamic objects?
- Limited real world evaluation. The paper claims to do over 300 evaluations across three different environments -- testing across a larger number of environments would provide more confidence in the results.
- The result of the ablation is not surprising at all: because the test metric used here is performance on the Indoor Navigation Dataset, the more of this exists in the training set, the smaller the error will be. A better ablation may have been perhaps the augmentation technique -- how much does the model suffer on the original test set, if the data augmentation is not done?

**Quality Of The Limitations Section:**

3

**Questions For Rebuttal:**

- The authors write: "The relatively small network size allows us to implement our policy at a higher frame rate onboard, which is a significant advantage over the baselines that try to run larger models on edge compute." The baselines were not necessarily designed for edge compute -- if the baselines were given the same number of attempts at inference, how would they perform? Is this difference only for dynamic objects?
- The distribution of the training objects has a long-tail -- what about the testing objects? How diverse are these? Furthermore, was something special done to tackle this challenge, e.g. balanced sampling during training?
- Where does the supervision for the velocity commands come from?
- Would this work across robot embodiments? If so, could this be demonstrated? Given the short height of the robot, to looks as though many viewpoints from the datasets (e.g. YouTube videos) may have been from an out-of-distribution viewpoint...
- What is the field of view of the camera? In the person-chasing demo in the video, seems like the camera needs to have a large vertical field of view to be able to see the orange jacket the person is wearing. On a related note, it looks like a fisheye camera is used on the robot -- how does the visual policy transfer to such a camera?
- The demonstrations in the dataset don't have to be optimal right? Wouldn't offline RL be better than behavior cloning for something like this? Could this be an additional baseline?
- Does the policy output a STOP action? If the supervision is coming from ego-centric non-robot videos, then even when the target object is reached, the ground truth velocity may not be zero, correct? In many of the videos, it appears as if the object either crashes into the target object, or the video is clipped before this can happen
- Is the capacity of the model the same as that of the baselines? Is it possible that the gain in performance is coming a better visual backbone (ResNet-FiLM)?

**Robotics Focus:**

4

**Summary Of Paper:**

This paper introduces LeLan, a methodology to leverage large-scale unlabelled egocentric data to improve language-conditioned object navigation. The method works by leveraging large VLMs. The authors demonstrate that the proposed methodology outperforms existing baselines.

**Summary Of Recommendation:**

The paper has many advantages and disadvantages and questions, but the disadvantages and questions seem to outweigh the advantages (see above).

---

### Author Rebuttal · Authors · 2024-08-10

We uploaded three items, 1) our revised paper, 2) video for collision avidance, and 3) cross embodiement evaluation with the quadruped robot, GO1.

---

### Decision · Program_Chairs · 2024-09-04

**Decision:**

Accept

**Comment:**

The authors propose a method for using large-scale, unlabeled data for last-mile robot navigation using VLMs. They compare it to a number of baselines, including a recent state of the art method, and show theirs is superior. They test in several diverse real-world environments. The approach itself seems straightforward and leverages large-scale offline data. The approach is clear, and the paper is solidly written.

The authors did a good job with their rebuttal, in particular making it clear that the baselines were stronger than it seemed, and adding more real-world experiments, as well as clarifying the paper's contributions to last-mile navigation.

Strengths:
- Using video from e.g. YouTube is a very powerful idea and it's good the authors were able to achieve this
- Thorough experiments and ablations
- Useful data for future research
- Good baselines

Weaknesses:
- Unclear how challenging the navigation tasks are - if the robots start very close to target objects it is not very interesting
- Needs more discussion of how much it cost and how well the method scales
- Limitations in data collection; can it handle obstacle avoidance, for example